# MedFuse: Multiplicative Embedding Fusion for Irregular Clinical Time Series

## Abstract

Clinical time series derived from electronic health records (EHRs) are inherently irregular, with asynchronous sampling, missing values, and heterogeneous feature dynamics. While numerical laboratory measurements are highly informative, existing embedding strategies usually combine feature identity and value embeddings through additive operations, which constrains their ability to capture value-dependent feature interactions. We propose MedFuse, a framework for irregular clinical time series centered on the MuFuse (Multiplicative Embedding Fusion) module. MuFuse fuses value and feature embeddings through multiplicative modulation, preserving feature-specific information while modeling higher-order dependencies across features. Experiments on three real-world datasets covering both intensive and chronic care show that MedFuse consistently outperforms state-of-the-art baselines on key predictive tasks. Analysis of the learned representations further demonstrates that multiplicative fusion enhances expressiveness and supports cross-dataset pretraining. These results establish MedFuse as a generalizable approach for modeling irregular clinical time series.

## 1 Introduction

Clinical time series from electronic health records (EHRs) are central to a wide range of predictive and monitoring tasks in healthcare, yet their irregular structure presents persistent modeling challenges. Unlike words in sentences or signals sampled at fixed intervals, clinical variables are measured on heterogeneous schedules with irregular gaps, leading to high missingness and asynchronous observations. For example, vital signs may be monitored frequently during hospitalization, while laboratory tests are ordered only when clinically indicated, and some patients may miss scheduled visits or follow-up tests altogether. Numerical features such as laboratory values are especially difficult to represent: they encode complex information in continuous ranges and, in principle, demand infinitely many representations. Effective models must therefore handle nonuniform sampling, sparsity, and diverse temporal dynamics across a large set of patient variables.

A growing line of work addresses these challenges by tokenizing each measurement as a (feature identity, value, timestamp) triplet (Tipirneni & Reddy, 2022b). This formulation allows models to learn directly from observed events and avoid explicit imputation, yielding an imputation-free workflow. The "each value as token" (EVAT) paradigm (Huang et al., 2024) naturally accommodates asynchronous sampling, since tokens are instantiated only when measurements occur. Within EVAT, most approaches combine feature, value, and time embeddings through additive composition (Li et al., 2020; Rasmy et al., 2021; Tipirneni & Reddy, 2022b). Additive fusion has been effective in practice: it integrates periodic time information (Vaswani et al., 2017b), treats values as learnable embedding offsets, and provides a simple yet scalable mechanism for modeling large vocabularies of clinical events. However, this design inherently constrains expressiveness. Treating the numerical value primarily as an additive shift to a base feature embedding limits the model's abil-

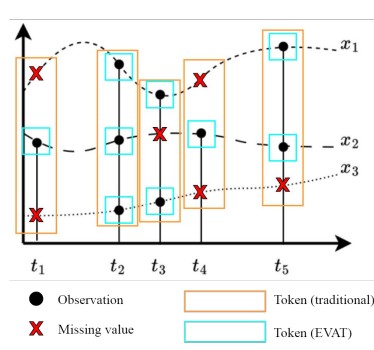

Figure 1: Illustration of EVAT.

ity to capture value-dependent, nonlinear interactions, such as how small versus large deviations in a laboratory test can imply qualitatively different clinical states. Consequently, additive fusion may struggle to represent context-sensitive effects critical for robust clinical prediction.

To address these limitations, we propose MedFuse, a framework for modeling irregular numerical clinical time series centered on a novel embedding module, MuFuse (Multiplicative Embedding Fusion). Instead of adding a value embedding to a feature vector, MuFuse performs value-conditioned multiplicative fusion, where the numerical measurement modulates the feature embedding through element-wise scaling. This yields token representations that vary nonlinearly with the observed value in a feature-specific manner. Multiplicative fusion has been shown to provide stronger semantic integration than addition or concatenation in other domains (Chrysos et al., 2025), and we demonstrate its effectiveness in the clinical setting. Importantly, we show that a recent state-of-the-art method for irregular EHR modeling (Huang et al., 2024) is in fact a special case of MuFuse, establishing our formulation as a more general fusion mechanism. By applying this principle to each (feature, value, timestamp) triplet, MuFuse preserves feature identity while allowing numerical values to shape the embedding in a nonlinear, feature-specific way, capturing clinically meaningful distinctions such as the difference between a slight increase and a sharp rise in creatinine.

We evaluate MedFuse on three real-world clinical datasets covering both intensive care and chronic disease settings. MedFuse consistently outperforms strong baselines, with ablations confirming the benefits of multiplicative fusion over additive schemes. Transfer experiments further show that learned feature embeddings can be reused across datasets with partially overlapping variables, supporting efficient pretraining and adaptation.

Our contribution can be summarized as follows:

1. **Multiplicative value–feature fusion.** We propose **MuFuse**, a novel embedding module that performs value-conditioned multiplicative fusion. This mechanism allows numerical values to modulate feature embeddings in a nonlinear, feature-specific way, enabling richer interactions without expanding the embedding vocabulary.

2. **A generalizable, imputation-free framework.** Built on MuFuse, **MedFuse** adopts the (feature, value, timestamp) *triplet* tokenization scheme to directly model irregular measurements without imputation. It unifies numerical and categorical events with efficient temporal encoding.

3. **Comprehensive validation and transferability.** MedFuse consistently outperforms strong baselines across intensive-care and chronic-disease datasets. Ablation studies highlight the advantage of multiplicative over additive fusion, and transfer experiments show that learned feature embeddings can be reused across datasets with partially overlapping variables.

## 2 RELATED WORK

Sequence models form the foundation of multivariate time-series (MTS) learning. Recurrent architectures such as Long Short-Term Memory (LSTM) (Hochreiter & Schmidhuber, 1997) and Gated Recurrent Unit (GRU) (Cho et al., 2014) established classic baselines, while attention-based Transformers extended this capacity to capture long-range dependencies (Vaswani et al., 2017a). To address irregular sampling and missing values, some methods integrate imputation into the learning process. For example, Bidirectional Recurrent Imputation for Time Series (BRITS) jointly learns imputations with recurrent prediction, reducing the train–test mismatch between filled and observed values (Cao et al., 2018). More recently, efficient Transformer variants such as Informer, which employs sparse self-attention, have demonstrated competitive accuracy–efficiency trade-offs for long-sequence forecasting (Zhou et al., 2021).

In healthcare, where EHR-derived time series are sparse and asynchronous, modeling pipelines have often relied on explicit imputation before outcome prediction. Simply Attend and Diagnose (SAnD) (Song et al., 2018a) eliminates recurrence with masked self-attention and positional encodings but requires resampling to a regular grid, which may blur feature-specific temporal signals. Imputation-free approaches bypass this issue. For instance, Multi-time Attention Networks (mTAN) employ continuous-time attention to reason directly over irregular observations (Shukla & Marlin, 2021a).

A parallel line of work represents each measurement as a token, avoiding missing-value imputation and preserving asynchrony by construction. This EVAT perspective has shown strong results: Self-supervised Transformer for Time-Series (STraTS) trains a Transformer on (feature, time, value) triplets with self-supervised objectives (Tipirneni & Reddy, 2022a), while Scalable Numerical Embeddings (SCANE) achieves state-of-the-art performance with a specialized value-scaling mechanism (Huang et al., 2024). However, the source of SCANE's advantage over additive schemes remains underexplored (see Appendix E).

Building on these advances, our work focuses on the unresolved challenge of fusing the components of each numerical EHR observation. Prior EVAT methods have relied mainly on additive or concatenative operations, which limit the ability to capture value-dependent, nonlinear feature interactions. In contrast, we introduce MuFuse, a value-conditioned multiplicative fusion module that modulates feature embeddings with observed values. MuFuse retains linear complexity in the embedding dimension, is compatible with standard sequence backbones such as the Transformer encoder (Vaswani et al., 2017a), and is designed to capture richer feature–value interactions under irregular sampling. The design of MuFuse and the overall MedFuse framework are detailed in the following section.

## 3 METHODOLOGY

### 3.1 PROBLEM SETUP AND NOTATION

Let $\mathcal{O} = \{(f, v, t)\}$ denote the set of observed triplets in a numerical multivariate time series, where $f \in \{1, \ldots, F\}$ indexes the feature identity (e.g., lab test type), $v \in \mathbb{R}$ is the recorded value, and $t \in \mathbb{R}_+$ is the corresponding (possibly overlapping) time converted from the original timestamp. We mask missing entries using an indicator $M_{f,t} \in \{0, 1\}$, and only tuples with $M_{f,t} = 1$ (i.e., actually observed records) are tokenized and passed to the downstream model.

Each observed triplet $(f, v, t)$ is mapped to three embeddings: a feature embedding $\mathbf{e}_f$, a value embedding $\mathbf{e}_v$, and a time embedding $\mathbf{e}_t$. These components are then combined through the proposed MuFuse module (Section 3.2) to form a unified token representation $\mathbf{e}_{f,v,t} \in \mathbb{R}^d$, which serves as input to downstream sequence modeling.

### 3.2 MUFUSE: MULTIPLICATIVE EMBEDDING FUSION

*MuFuse* is the core module of MedFuse. It integrates the feature, value, and time embeddings of each triplet using value-conditioned multiplicative modulation, enabling richer feature–value interactions than additive or concatenative schemes. We next describe how each component is embedded and how the fusion is performed.

#### 3.2.1 FEATURE IDENTITY EMBEDDING

Each feature type $f \in \{1, \ldots, F\}$ is associated with a unique learnable embedding vector

$$\mathbf{e}_f \in \mathbb{R}^d, \tag{1}$$

obtained through a standard lookup table. This embedding provides a stable representation of the feature identity.

#### 3.2.2 VALUE EMBEDDING

To represent the observed scalar $v$, we employ a shared nonlinear projector $\phi : \mathbb{R} \to \mathbb{R}^{d'}$:

$$\mathbf{z}_v = \phi(v) \in \mathbb{R}^{d'}. \tag{2}$$

To capture feature-specific variations (e.g., different scales across lab tests), we further apply a learnable affine transformation conditioned on the feature type $f$:

$$\mathbf{e}_{v|f} = \boldsymbol{\gamma}_f \odot \mathbf{z}_v + \boldsymbol{\beta}_f \in \mathbb{R}^{d'}, \tag{3}$$

where $\boldsymbol{\gamma}_f, \boldsymbol{\beta}_f \in \mathbb{R}^{d'}$ are feature-specific parameters and $\odot$ denotes element-wise multiplication. For simplicity, in the rest of this article, we denote $\mathbf{e}_{v|f}$ as $\mathbf{e}_v$.

### 3.2.3 MULTIPLICATIVE FUSION

In this step, we aim to integrate the feature identity embedding $\mathbf{e}_f \in \mathbb{R}^d$ and the value embedding $\mathbf{e}_v \in \mathbb{R}^{d'}$. We *extended* the standard Hadamard product as our operation of the proposed multiplicative fusion, as it was found to be efficient in aggregating information for deep models (Chrysos et al., 2025). When $d' = d$, the proposed multiplicative fusion MuFuse can be expressed using the standard Hadamard product between the two involved embeddings (vectors):

$$\text{MuFuse}\left(\mathbf{e}_f, \mathbf{e}_v\right) \; = \; \mathbf{e}_f \odot \mathbf{e}_v \; = \; \mathbf{e}_{f,v} \tag{4}$$

However, in real-world applications, the best practice for embedding dimensions for feature identity and observed value can differ ($d' \neq d$). To address this issue, we generalize the multiplicative fusion through *entry broadcasting*. To illustrate this, without loss of generality, we assume $d > d'$ and $d = d' \times k, k \in \mathbb{N}$. To match the dimensions, we repeat each entry of $\mathbf{e}_v$ for $k$ times to generate an extended embedding $\mathbf{e}_{v'} \in \mathbb{R}^d$. All entries will pass through a sigmoid function to suppress abnormal values. Then, the fusion can be expressed again using the standard Hadamard product between $\mathbf{e}_f$ and $\mathbf{e}_{v'}$.

An alternative expression to accentuate the interaction between $\mathbf{e}_f$ and $\mathbf{e}_v$ is to first partition the feature identity embedding into $k$ contiguous blocks,

$$\mathbf{e}_f \; = \; \left[\mathbf{e}_f^{(1)}; \mathbf{e}_f^{(2)}; \ldots; \mathbf{e}_f^{(d/k)}\right], \quad \mathbf{e}_f^{(i)} \in \mathbb{R}^{d/k}. \tag{5}$$

Then, we compute the entries $v^j$ ($j = 1, 2, ..., d/k$) of $\mathbf{e}_v$ as *per-block gates* by a bounded nonlinear function $g$. As depicted earlier, we choose $g$ as the sigmoid function $\sigma$.

$$g(v^j) \; = \; \sigma(v^j) \in (0, 1). \tag{6}$$

Finally, we apply each scalar "gate" to its corresponding block to conduct the scalar multiplication:

$$\mathbf{e}_{f,v}^{(i)} \; = \; g(v^j)\,\mathbf{e}_f^{(i)}, \qquad \mathbf{e}_{f,v} \; = \; \left[\mathbf{e}_{f,v}^{(1)}; \ldots; \mathbf{e}_{f,v}^{(d/k)}\right] \in \mathbb{R}^d, \qquad j = i. \tag{7}$$

Value embeddings in MuFuse act as modulators of feature identity embeddings, enabling expressive modeling of complex feature–value interactions. Conversely, the feature identity embedding can regulate how value effects are expressed through dimensionality choices and gating configuration. The relative sizes of $d$ and $d'$ thus provide a flexible design space that can be tuned to the task, balancing representational capacity with efficiency.

### 3.3 EMBEDDING FOR CATEGORICAL FEATURES

Since categorical embedding is not the main focus of this work, we adopt a straightforward approach. For a categorical observation, we concatenate its feature identity embedding $\mathbf{e}_f \in \mathbb{R}^d$ with the class lookup embedding $\mathbf{e}_c \in \mathbb{R}^{d_c}$, and apply a linear transformation to obtain the final embedding:

$$\mathbf{e}_{f,c} \; = \; W_{\text{cat}}\, Concat\left(\mathbf{e}_f, \mathbf{e}_c\right) \; \in \; \mathbb{R}^d, \tag{8}$$

where $Concat$ denotes concatenation and $W_{\text{cat}} \in \mathbb{R}^{d \times (d + d_c)}$.

### 3.4 TIME EMBEDDING

We adopt the classic sinusoidal positional encoding. Let $t$ denote the elapsed time converted from a token's timestamp. A $d$-dimensional sinusoidal vector $\mathbf{p}_t$ is formed by interleaving sines and cosines with geometrically spaced wavelengths $\omega_i$:

$$\mathbf{p}_t[2i] = \sin\left(t/\omega_i\right), \qquad \mathbf{p}_t[2i+1] = \cos\left(t/\omega_i\right), \quad i = 0, \ldots, \tfrac{d}{2} - 1. \tag{9}$$

For embeddings of any token, including fused numerical feature embedding $\mathbf{e}_{f,v}$ in Section 3.2.3 and categorical embedding in Section 3.3, temporal information is injected by addition instead of multiplicative fusion:

$$\mathbf{e}_{f,v,t} \; = \; \mathbf{e}_{f,v} \;(\text{or } \mathbf{e}_{f,c}) + \mathbf{p}_t \; \in \; \mathbb{R}^d. \tag{10}$$

Same $\mathbf{p}_t$ is broadcast to all $\mathbf{e}_{f,v}$ with time $t$. This preserves a clean separation between content $\mathbf{e}_{f,v}$ and temporal pattern $\mathbf{p}_t$. In Appendix F, we will discuss why addition is more suitable than multiplicative fusion for temporal information.

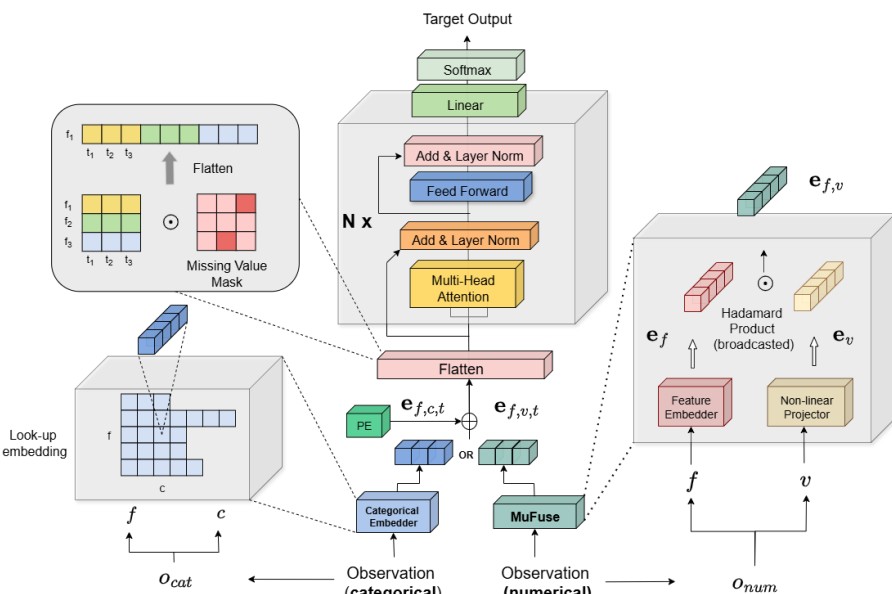

Figure 2: **MedFuse architecture with MuFuse value–feature fusion.** Numerical observations $(f, v, t)$ are embedded by MuFuse: a feature embedder maps the identity $f \to \mathbf{e}_f \in \mathbb{R}^d$, a nonlinear projector maps the measured value $v \to \mathbf{e}_v \in \mathbb{R}^d$, and the token embedding is formed by element-wise (broadcasted) Hadamard product $\mathbf{e}_{f,v} = \mathbf{e}_f \odot \mathbf{e}_v$, followed by adding a time/positional encoding to yield $\mathbf{e}_{f,v,t}$. Categorical events use a categorical embedder to produce $\mathbf{e}_{f,c,t}$. All tokens are flattened into a sequence with a missing-value mask and processed by an $N$-layer Transformer encoder; a linear softmax head outputs the target distribution. By modulating $\mathbf{e}_f$ multiplicatively with the observed value, MuFuse preserves feature identity while enabling rich, value-dependent interactions.

### 3.5 COMPLETE ARCHITECTURE (MEDFUSE)

To address downstream tasks such as risk prediction, we integrate MuFuse as the embedding layer with a standard *Transformer* encoder (Vaswani et al., 2017b), forming the overall MedFuse framework. Figure 2 illustrates the complete architecture: observed triplets are first mapped into embeddings via MuFuse, combined with time encodings, masking out the token belonging to missing values, and then processed by the *Transformer* encoder (Vaswani et al., 2017b) to generate sequence representations for classification.

## 4 EXPERIMENT AND RESULT

### 4.1 DATASETS

We evaluate MedFuse on three irregular, high-missingness clinical time-series cohorts widely used in healthcare modeling: the PhysioNet 2012 ICU mortality dataset (P12) (Silva et al., 2012), the MIMIC-III ICU benchmark (MI3) (Johnson et al., 2016), and a private longitudinal hepatocellular carcinoma (HCC) cohort from a medical center. A consistent preprocessing protocol is applied across datasets: raw measurements are grouped into fixed windows (ICU tasks: 2-hour windows over a 48-hour horizon; HCC: 90-day windows over a 1-year horizon), with numeric variables summarized by median and categorical variables by the most frequently observed class. We adopt an imputation-free, event-level tokenization in which only observed measurements generate tokens, while missing entries remain unfilled. The full feature inventory (variable definitions, types, and windowing rules) and dataset statistics are provided in Appendix B.

## 4.2 STUDY DESIGN

**ICU mortality (P12, MI3).** For the PhysioNet 2012 and MIMIC-III ICU cohorts, we use an observation window of 48 hours with 2-hour summarization. The task is to predict *in-hospital mortality* after the observation window during the hospital stay: $label = 1$ if the patient dies in the hospital, and $label = 0$ otherwise. This binary classification setup follows standard ICU benchmarks, which use 48-hour histories for outcome prediction.

**HCC onset risk.** For the longitudinal HCC cohort, we use a one-year observation window ($L = 1$ year) with 90-day summarization. $t_0$ denotes the entry date. The task is to predict long-term onset risk: given one year of history, estimate the probability of developing HCC within a future horizon $\tau$. The $label = 1$ if a diagnosis occurs in $[t_0+L, t_0+L+\tau)$, and $label = 0$ otherwise. In our main experiments, we set $\tau = 5$ years, reflecting clinical surveillance practice where near-term history informs multi-year risk.

## 4.3 BASELINES AND BENCHMARKS

We compare MedFuse with representative sequence models for multivariate time series, with emphasis on methods evaluated in clinical EHR contexts. Classical ensemble methods include **Random Forest** and **XGBoost** (Chen & Guestrin, 2016), which remain strong baselines for structured medical data. Among deep learning approaches, we consider a standard **Transformer** encoder (Vaswani et al., 2017b) as a general attention-based model, and **TCN** (Bai et al., 2018), which captures sequential dependencies using dilated causal convolutions. We also evaluate architectures designed specifically for irregular clinical time series: **SAnD** (Song et al., 2018b), which adapts attention with masking strategies; **mTAN** (Shukla & Marlin, 2021b), which employs continuous-time attention to handle irregular sampling; and **STraTS** (Tipirneni & Reddy, 2022b), which encodes each measurement as a (feature, time, value) triplet and learns event-level representations via self-supervision. Finally, we include **SUMMIT** (Huang et al., 2024), which introduces scalable numerical embeddings to modulate feature vectors with observed values and has reported state-of-the-art results on irregular EHR modeling.

## 4.4 EVALUATION METRICS

Because all datasets are imbalanced, we report the area under the precision–recall curve (AUPRC) (Saito & Rehmsmeier, 2015) as the primary evaluation metric, as it better captures performance under skewed class distributions. For completeness, we also report the area under the receiver operating characteristic curve (AUROC) and the concordance index (c-index). For P12 and MI3, where event times are unavailable, the c-index is replaced with accuracy using a fixed threshold of 0.5.

## 4.5 PERFORMANCE COMPARISON

Table 1 presents the performance of MedFuse and all baselines on MI3, P12, and HCC. For each metric, the best score is shown in **bold** and the second-best is underlined. Values with $\pm$ indicate 95% confidence intervals, estimated via 1000 bootstrap samples.

MedFuse achieves the highest AUPRC on all three datasets, establishing new state-of-the-art performance on the primary evaluation metric. It also obtains the best AUROC and accuracy on MI3 and competitive auxiliary results on P12 and HCC. These outcomes indicate that multiplicative fusion enables MedFuse to capture value–feature interactions more effectively than additive schemes, delivering robust improvements under irregular and high-missingness conditions. Overall, MedFuse consistently outperforms SUMMIT and other strong baselines, underscoring its effectiveness for modeling irregular clinical time series.

## 4.6 ABLATION STUDY

To assess the impact of our fusion design, we compare MuFuse with additive and concatenative schemes. Table 2 reports results on P12, showing clear gains from multiplicative fusion. Similar trends are observed on MI3 and HCC, with detailed results provided in Appendix G. These findings

Table 1: Performance comparison.

| Dataset | MI3 | | | P12 | | | HCC | | |
|---|---|---|---|---|---|---|---|---|---|
| Metric | AUPRC | AUROC | Accuracy | AUPRC | AUROC | Accuracy | AUPRC | AUROC | c-index |
| Random Forest | 0.4367 ±0.0517 | 0.8319 ±0.0209 | 0.8965 ±0.0105 | 0.4805 ±0.0533 | 0.8270 ±0.0228 | 0.8663 ±0.0146 | 0.3934 ±0.0583 | 0.8705 ±0.0232 | 0.8637 ±0.0227 |
| XGBoost | 0.4553 ±0.0527 | 0.8247 ±0.0209 | 0.8968 ±0.0105 | 0.4980 ±0.0544 | 0.8453 ±0.0203 | 0.8708 ±0.0140 | 0.3887 ±0.0592 | 0.8714 ±0.0215 | 0.8644 ±0.0209 |
| Transformer Enc. | 0.5074 ±0.0510 | 0.8606 ±0.0187 | 0.8953 ±0.0105 | 0.5435 ±0.0560 | 0.8572 ±0.0200 | 0.8767 ±0.0131 | 0.4139 ±0.0571 | 0.8964 ±0.0171 | 0.8888 ±0.0171 |
| TCN | 0.5128 ±0.0377 | 0.8734 ±0.0165 | 0.8999 ±0.0098 | 0.4725 ±0.0494 | 0.8272 ±0.0263 | 0.8581 ±0.0134 | 0.3725 ±0.0661 | 0.8684 ±0.0493 | 0.8616 ±0.0187 |
| SAnD | 0.5463 ±0.0462 | 0.8774 ±0.0096 | 0.9023 ±0.0123 | 0.4615 ±0.0598 | 0.8227 ±0.0245 | 0.8674 ±0.0179 | 0.3769 ±0.0337 | 0.8836 ±0.0090 | 0.8763 ±0.0087 |
| mTAN | 0.5536 ±0.0359 | 0.8826 ±0.0163 | 0.9037 ±0.0227 | 0.4991 ±0.0521 | 0.8444 ±0.0267 | **0.8863** ±0.0127 | 0.4545 ±0.0264 | 0.8762 ±0.0135 | 0.8466 ±0.0138 |
| STraTS | 0.5886 ±0.0546 | 0.8936 ±0.0021 | 0.9044 ±0.0104 | 0.5206 ±0.0534 | 0.8596 ±0.0224 | 0.8253 ±0.0135 | 0.4270 ±0.0186 | 0.8963 ±0.0088 | 0.8888 ±0.0086 |
| SUMMIT | 0.6328 ±0.0277 | 0.9035 ±0.0092 | 0.9111 ±0.0060 | 0.5504 ±0.0563 | 0.8602 ±0.0197 | 0.8783 ±0.0129 | 0.4553 ±0.0577 | 0.8943 ±0.0179 | 0.8867 ±0.0179 |
| **MedFuse** | **0.6574** ±0.0270 | **0.9078** ±0.0087 | **0.9153** ±0.0058 | **0.5612** ±0.0558 | **0.8686** ±0.0190 | 0.8837 ±0.0558 | **0.4595** ±0.0556 | **0.9062** ±0.0163 | **0.8982** ±0.0158 |

Table 2: Ablation study of feature–value fusion strategies.

| PhysioNet 2012 | | | |
|---|---|---|---|
| Method | AUPRC | AUROC | Accuracy |
| MuFuse (ours) | **0.5612 ± 0.0558** | **0.8686 ± 0.0190** | **0.8837 ± 0.0558** |
| Adding | 0.5317 ± 0.0546 | 0.8549 ± 0.0205 | 0.8754 ± 0.0131 |
| Concatenate | 0.5291 ± 0.0564 | 0.8518 ± 0.0204 | 0.8779 ± 0.0129 |

confirm that value-conditioned multiplicative modulation captures feature–value interactions that additive schemes fail to represent, leading to more expressive and robust embeddings.

## 4.7 EFFECT OF VALUE EMBEDDING DIMENSION

As described in Section 3.2.3, the value embedding dimension is defined as $d' = d/k$ under a fixed feature identity embedding $\mathbf{e}_f \in \mathbb{R}^d$. To examine its impact, we fix $d = 144$ and vary the partitioning factor $k$. Figure 3 shows results on P12, where performance peaks at intermediate values of $k$, indicating that neither too coarse nor too fine partitioning is optimal. Similar trends are observed on MI3 and HCC (see Appendix H). This supports the design choice of MuFuse, where the broadcasted Hadamard product enables flexible alignment between feature and value embedding dimensions. Intuitively, intermediate dimensionality provides a balance between under-parameterizing value effects and overfitting feature–value interactions.

## 4.8 CROSS-DATASET ADAPTATION OF FEATURE IDENTITY EMBEDDINGS

We investigate whether the feature identity embeddings learned by MedFuse capture reusable, cohort-agnostic semantics by transferring only the feature embeddings for the overlapping feature set $\mathcal{F}_\cap$ (see Appendix C) between the two ICU cohorts (P12 and MI3). For each transfer direction, we follow this protocol: (i) Pre-train on the source cohort, (ii) Initialize the target model with the source-trained identity embeddings for $\mathcal{F}_\cap$, (iii) Keep all other settings identical to the from-scratch baseline (including initialization, hyperparameters, and seed), and (iv) Warm up the adaptation by briefly freezing the transferred embeddings before joint fine-tuning. This controlled setup isolates

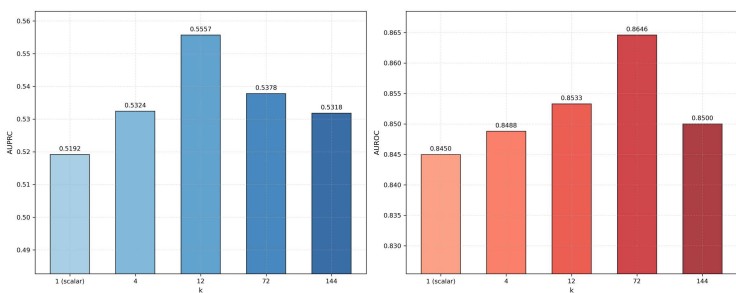

Figure 3: Comparison of different partitioning factors $k$ on P12.

the contribution of cross-cohort embedding transfer, enabling us to evaluate whether feature identity embeddings generalize across datasets.

Table 3: Cross-dataset transfer on ICU cohorts.

**P12 → MIMIC-III**

| Model | AUPRC | AUROC | Accuracy |
|---|---|---|---|
| MedFuse (from scratch) | **0.663872** (0.0276) | **0.911360** (0.0082) | **0.917436** |
| MedFuse + pretrained embeddings | 0.642217 (0.0269) | 0.902341 (0.0085) | 0.916310 |

**MIMIC-III → P12**

| Model | AUPRC | AUROC | Accuracy |
|---|---|---|---|
| MedFuse (sub-sample) | 0.527648 (0.0556) | 0.850109 (0.0202) | 0.879583 |
| MedFuse (from scratch) | 0.536134 (0.0586) | 0.853872 (0.0201) | 0.876250 |
| MedFuse + pretrained embeddings | **0.545436** (0.0568) | **0.855070** (0.0206) | **0.881250** |

As shown in Table 3, transferring from the larger source (MI3) to the smaller target (P12) yields consistent improvements, while the reverse direction (P12→MI3) leads to negligible or slightly negative transfer. To disentangle dataset identity from size, we also pretrain on a P12-sized *subsample* of MI3 before transferring to P12; this likewise results in slight negative transfer. These findings indicate that the benefit of cross-cohort adaptation is driven primarily by the scale of the source dataset rather than its identity, highlighting sample size as the key factor for learning reusable feature-level semantics.

## 5  DISCUSSION

### 5.1  ADVANTAGES OF MEDFUSE OVER EXISTING BENCHMARKS

Across both ICU mortality prediction (PhysioNet 2012, MIMIC-III) and long-term chronic disease prognosis (HCC cohort), MedFuse achieves higher AUPRC and AUROC under severe class imbalance, while remaining robust across heterogeneous cohorts. Traditional ensemble learners such as Random Forests and XGBoost remain competitive but rely on population-level imputation, which introduces bias and limits generalization under high missingness. Deep sequential models (e.g., recurrent and convolutional architectures) capture temporal patterns yet struggle with irregular sampling and long-range dependencies. Attention-based approaches, including SAnD, mTAN, STraTS, and SUMMIT, mitigate some of these challenges through imputation-free representations but largely rely on additive fusion of feature, value, and time embeddings. This additive design restricts the ability to model nonlinear feature–value interactions that are clinically meaningful. Our ablations confirm that MedFuse's gains arise from the *multiplicative* MuFuse module, which produces richer

token representations by conditioning feature embeddings on observed values. These findings position MedFuse not only as a strong alternative to existing models but also as a general framework for advancing the modeling of irregular clinical time series.

## 5.2 CLINICAL RATIONALE FOR MULTIPLICATIVE FUSION

Beyond the general benefits highlighted by Chrysos et al. (2025), we argue that the Hadamard product in MuFuse is particularly well-suited for modeling numerical EHR features due to domain–specific considerations. In this section, we will investigate it based on MuFuse's **Masking and collapse effects**.

In clinical contexts, different value ranges of a feature may correspond to the same risk phenotype. For example, both *hyponatremia* and *hypernatremia* can be associated with neurological symptoms such as seizures and altered mental status, although one corresponds to low sodium and the other to high sodium levels. With additive fusion, capturing such equivalence is difficult: it would require assigning identical embeddings to different value ranges, which removes flexibility and erases other meaningful distinctions. By contrast, the broadcasted Hadamard product in MuFuse allows a *masking effect*: even if the value embeddings differ, element-wise multiplication with the same feature embedding as an entry-level mask can collapse the two different embeddings into the same (i.e., a common representation). This mechanism naturally models medical equifinality, where different abnormal deviations (e.g., too low or too high of a laboratory measurement) can correspond to the same clinical risk. We provide an example in Appendix J to further demonstrate the property.

## 5.3 FINAL REMARK OF THE MULTIPLICATIVE FUSION

Recall from equation 4 that:

$$\text{MuFuse}\left(\mathbf{e}_f, \mathbf{e}_v\right) \ = \ \mathbf{e}_f \odot \mathbf{e}_v \ = \ \mathbf{e}_{f,v},$$

where $\mathbf{e}_v$ is a learnable value embedding.

This can be rewritten as:

$$\text{MuFuse}\left(\mathbf{e}_f, \mathbf{e}_v\right) \ = \ \mathbf{e}_f \odot \mathbf{e}_v = \mathbf{e}_f \odot \left(1 + \mathbf{e}'_v\right) = \mathbf{e}_f + \mathbf{e}_f \odot \mathbf{e}'_v, \tag{11}$$

where $1 + \mathbf{e}'_v$ is simply another parameterization of the value embedding $\mathbf{e}_v$. In contrast, additive fusion takes the form $\mathbf{e}_{f,v} = \mathbf{e}_f + \mathbf{e}_v$, where the value embedding contributes as an independent term, uninfluenced by $\mathbf{e}_f$. MuFuse instead introduces an explicit *interaction term*, $\mathbf{e}_f \odot \mathbf{e}'_v$, in which the modulation of $\mathbf{e}_v$ is conditioned on the feature identity $\mathbf{e}_f$. This interaction allows MuFuse to capture more expressive and feature-dependent value effects than additive fusion.

## 6 CONCLUSION

We presented **MedFuse**, a general framework for modeling irregular clinical time series. At its core is the **MuFuse** module, which performs value-conditioned multiplicative fusion between feature identity embeddings and numerical values. This design enables nonlinear, feature-specific modulation that more faithfully captures the semantics of clinical measurements than additive or concatenative schemes, while remaining efficient and imputation-free.

Evaluations on three real-world datasets—two intensive care cohorts and one chronic disease cohort—demonstrate that MedFuse consistently outperforms strong baselines across multiple predictive tasks. Ablation studies confirm the unique contribution of multiplicative fusion, and transfer experiments show that learned feature identity embeddings support cross-dataset adaptation, highlighting the potential for pretraining in heterogeneous healthcare settings.

Overall, MedFuse establishes multiplicative embedding fusion as a powerful paradigm for learning from irregular, high-missingness EHR data. Promising directions for future work include scaling MedFuse to large multimodal corpora, enhancing interpretability through causal or counterfactual reasoning, and integrating the framework into trustworthy clinical decision-support systems capable of operating in diverse real-world environments.

## REPRODUCIBILITY STATEMENT

We provide an anonymized code repository with all scripts to reproduce our results at anonymous.4open.science/r/MedFuse-19CE. It contains our method and all baselines.

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

APPENDIX

# Table of Contents

## A  HYPERPARAMETERS

Tables 4 to 6 list the hyperparameters of all models used to conduct our experiments. For our proposed MedFuse, we used Optuna (Akiba et al., 2019) for tuning on the validation set (split from the training set). For baseline models, we used the optimal hyperparameters reported in their original papers.

Table 4: Hyperparameter settings for the P12 dataset.

| Hyperparameter | TCN | SAnD | mTAN | STraTS | SUMMIT | MedFuse (ours) |
|---|---|---|---|---|---|---|
| $d_{model}$ | – | – | 256 | – | 144 | 144 |
| $ff_{dim}$ | – | – | 20 | – | 144 | 144 |
| hidden_size | 64 | 64 | 64 | 64 | – | – |
| num_layer | 4 | 4 | 1 | 2 | 1 | 32 |
| learning rate | 5e-4 | 5e-4 | 5e-4 | 5e-4 | 3e-5 | 3e-5 |
| early stopping (epoch) | 23 | 30 | 250 | 50 | 350 | 30 |

## B  DATASET STATISTICS

Tables 7 to 9 list the characteristics of each dataset used to conduct our experiments.

Table 5: Hyperparameter settings for the MIMIC-III dataset.

| Hyperparameter | TCN | SAnD | mTAN | STraTS | SUMMIT | MedFuse (ours) |
|---|---|---|---|---|---|---|
| $d_{model}$ | – | – | 256 | – | 144 | 144 |
| $ff_{dim}$ | – | – | 20 | – | 80 | 80 |
| hidden_size | 128 | 64 | 64 | 64 | – | – |
| num_layer | 4 | 4 | 1 | 2 | 1 | 32 |
| learning rate | 5e-4 | 5e-4 | 5e-5 | 5e-4 | 3e-5 | 3e-5 |
| early stopping (epoch) | 23 | 25 | 200 | 50 | 380 | 30 |

Table 6: Hyperparameter settings for the HCC dataset.

| Hyperparameter | TCN | SAnD | mTAN | STraTS | SUMMIT | MedFuse (ours) |
|---|---|---|---|---|---|---|
| $d_{model}$ | – | – | 256 | – | 144 | 144 |
| $ff_{dim}$ | – | – | 20 | – | 144 | 144 |
| hidden_size | 64 | 64 | 64 | 100 | – | – |
| num_layer | 4 | 4 | 1 | 2 | 8 | 32 |
| learning rate | 5e-4 | 5e-4 | 1e-4 | 5e-4 | 3e-5 | 3e-5 |
| early stopping (epoch) | 75 | 29 | 54 | 44 | 100 | 50 |

Table 7: **Summary of the PhysioNet 2012 dataset.**

| Statistic | Value |
|---|---|
| Number of numerical variables | 40 |
| Number of categorical variables | 2 |
| Number of patient stays (samples) | 11,988 |
| Number of positive cases (mortality) | 1,707 |
| Number of negative cases (survivors) | 10,281 |
| Class imbalance ratio (positive rate) | 0.142 |
| Average missing rate (after summarization) | 0.7377 |
| Observation window length | 48 hours |
| Summarization window length | 2 hours |
| Number of timestamps after summarization | 24 |

Table 8: **Summary of the MIMIC-III dataset.**

| Statistic | Value |
|---|---|
| Number of numerical variables | 128 |
| Number of categorical variables | 4 |
| Number of patient stays (samples) | 52,871 |
| Number of positive cases (mortality) | 6,506 |
| Number of negative cases (survivors) | 46,365 |
| Class imbalance ratio (positive rate) | 0.140 |
| Average missing rate (after summarization) | 0.8814 |
| Observation window length | 48 hours |
| Summarization window length | 2 hours |
| Number of timestamps after summarization | 24 |

Table 9: **Summary of the HCC dataset.**

| Statistic | Value |
|---|---|
| Number of numerical variables | 30 |
| Number of categorical variables | 8 |
| Number of patient records (samples) | 34,296 |
| Number of positive cases (develop HCC) | 1,523 |
| Number of negative cases (no HCC) | 32,773 |
| Class imbalance ratio (positive rate) | 0.046 |
| Average missing rate (after summarization) | 0.7464 |
| Observation window length | 1 year |
| Summarization window length | 90 days |
| Number of timestamps after summarization | 4 |

## C  LIST OF THE OVERLAPPING FEATURES BETWEEN P12 AND MI3

Table 10 lists the overlapping features between P12 and MI3 in the pre-training-adaptation experiment. Overlapping proportion: $59.5\%$ for P12 (25 out of 42 features) and $18.9\%$ for MI3 (25 out of 132 features).

Table 10: Overlapping features between PhysioNet 2012 and MIMIC-III, grouped by category.

| Category | Feature |
|---|---|
| **Vitals & Anthropometrics** | Age |
| | Height |
| | Weight |
| | Heart Rate |
| | Temperature |
| | Respiratory Rate |
| | Oxygen Saturation |
| | Systolic Blood Pressure |
| | Diastolic Blood Pressure |
| | Mean Arterial Pressure |
| **Lab Tests** | Alanine Aminotransferase (ALT) |
| | Aspartate Aminotransferase (AST) |
| | Alkaline Phosphatase (ALP) |
| | Albumin |
| | Total Bilirubin |
| | Creatinine (Serum) |
| | Blood Urea Nitrogen (BUN) |
| | Sodium |
| | White Blood Cell Count |
| | Platelet Count |
| | Glucose (Serum) |
| | Lactate |
| | Blood pH |
| **Other Parameters** | Fraction of Inspired Oxygen (FiO$_2$) |
| | Urine Output |

## D EMBEDDING PATTERN OF MEDFUSE

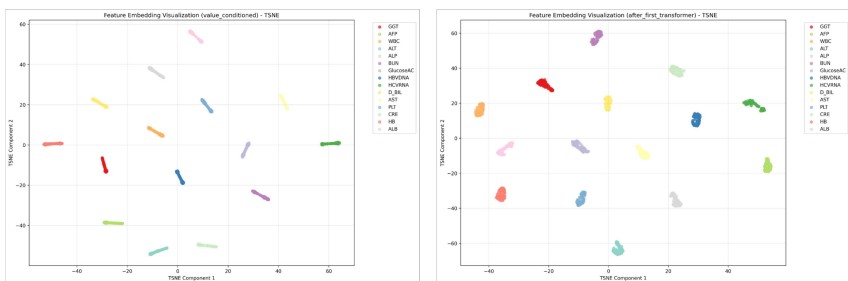

Figure 4: TSNE (Maaten & Hinton, 2008) visualization of MedFuse's $\mathbf{e}_{f,v}$ on the HCC dataset before (*Left*) and after (*Right*) passing through the first layer of the *Transformer* encoder. Each point represents a token's embedding, colored by its feature type. We can clearly see that MedFuse successfully embeds tokens with the same feature type into a cluster. This characteristic is preserved after passing the embedding through the first layer of the *Transformer* encoder, demonstrating MedFuse's robustness.

## E   RELATIONSHIP BETWEEN MEDFUSE AND PREVIOUS SOTA

Huang et al. (2024) proposed SCANE (and the corresponding classifier *SUMMIT*), achieving second-best performance among all tested models. It also adopts the EVAT strategy, directly applying scalar multiplication of the observed numerical EHR value to the feature identity embedding, outperforming its additive fusion counterpart (Tipirneni & Reddy, 2022b). We argue that the direct value multiplication of SCANE is actually a special case of MuFuse with $d' = 1$, while discarding any further transformation on the observed value. This observation may justify SCANE's advantages on EHR datasets based on our analyses of how multiplicative fusion benefits numerical EHR modeling throughout this paper. Meanwhile, MuFuse generalizes SCANE's mechanism with a more flexible architecture to achieve the sweet spot of dimension assignment (see Section 4.7), with the underlying rationale being more clearly investigated.

## F   WHY MULTIPLICATIVE FUSION DOES NOT WORK FOR TIME EMBEDDINGS?

Finally, we would like to discuss why MuFuse was not applied to time embeddings. In our additional experiment, adding the time embedding to $\mathbf{e}_{f,v}$ performed better than multiplicative fusion, as shown in Table 11. All variants use the same experimental settings (and identical hyperparameters). In the second row, "$\odot$" denotes the broadcasted Hadamard product used in MuFuse.

Table 11: Temporal embedding fusions on MIMIC-III.

| Model variant | AUPRC | AUROC | Accuracy |
|---|---|---|---|
| MedFuse + sinusoidal PE (additive fusion) | **0.6717** | **0.9148** | **0.9176** |
| MedFuse $\odot$ time PE (MuFuse) | 0.6495 | 0.9089 | 0.9159 |

To explore the underlying cause of this phenomenon, we investigated the first five dimensions of the time embedding after they were fused to the feature identity embedding through addition and MuFuse, respectively, as shown in Figure 5:

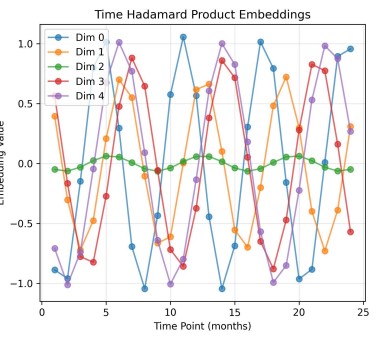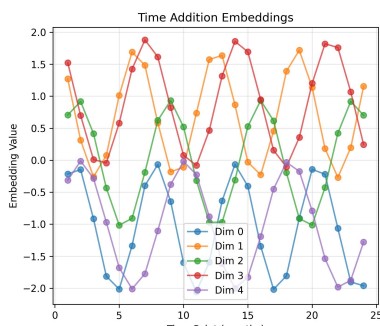

Figure 5: Comparison of the time embedding fusion effect. (*Left*): Fused to the feature identity embedding through MuFuse (broadcasted Hadamard product). (*Right*): Fused to the feature identity embedding through addition.

In Figure 5, we can observe that the time embedding is just shifted by the feature identity embedding (serving as a DC signal), preserving its informative AC signal magnitude (and thus the major spectrum pattern). On the contrary, multiplicative fusion changes the AC magnitude and the spectral composition ratio, thus breaking the regular representation pattern of the original sinusoidal time embedding. While orderly positional encoding has been shown to be crucial for sequential data modeling (He et al., 2020; Su et al., 2024), we argue that, unlike the numerical EHR case, addition is more suitable than the multiplicative operator for temporal information fusion.

## G   MORE ON THE ABLATION STUDY

We report the ablation study results on the other two datasets in Table G. Multiplicative fusion (Mu-Fuse) consistently demonstrates its advantage over addition and concatenation on different datasets.

Table 12: Ablation study of feature–value fusion strategies. Best results per dataset are **bold**.

| **MIMIC-III** | | | |
| --- | --- | --- | --- |
| Method | Accuracy | AUROC | AUPRC |
| MuFuse (ours) | **0.9177 ± 0.0055** | **0.9148 ± 0.0080** | **0.6717 ± 0.0283** |
| Adding | 0.9143 ± 0.0056 | 0.9128 ± 0.0084 | 0.6633 ± 0.0269 |
| Concatenate | 0.9162 ± 0.0055 | 0.9147 ± 0.0081 | 0.6671 ± 0.0277 |

| **HCC** | | | | |
| --- | --- | --- | --- | --- |
| Method | Accuracy | AUROC | AUPRC | c-index |
| MuFuse (ours) | **0.9593 ± 0.0044** | **0.9062 ± 0.0163** | **0.4595 ± 0.0556** | **0.8982 ± 0.0158** |
| Adding | 0.9570 ± 0.0047 | 0.9054 ± 0.0157 | 0.4353 ± 0.0562 | 0.8976 ± 0.0155 |
| Concatenate | 0.9593 ± 0.0044 | 0.9020 ± 0.0161 | 0.4215 ± 0.0578 | 0.8941 ± 0.0160 |

## H   MORE ON THE PARTITIONING FACTORS $k$ TESTS

We report the results of different partitioning factors on the other two datasets in Figures 6 and 7. We can observe that the optimal factor varies among datasets/medical tasks, consistent with the result on P12 reported in the main text.

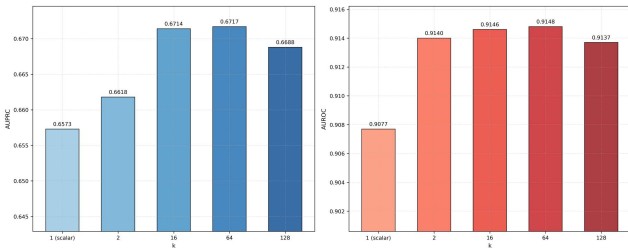

Figure 6: Comparison of different partitioning factors $k$ on MI3.

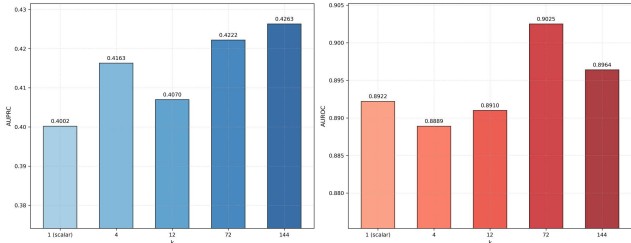

Figure 7: Comparison of different partitioning factors $k$ on HCC.

# I LLM USAGE

We used ChatGPT to polish our English writing in all paragraphs of this article.

# J EXAMPLE OF HOW MUFUSE FITS THE MEDICAL EQUIFINALITY NATURE

Suppose two observations of hypokalemia (low potassium in blood) and hyperkalemia (high potassium in blood) derive different value embeddings $\mathbf{e}_{v(low)}$ and $\mathbf{e}_{v(high)}$ while share the same feature identity embedding $\mathbf{e}_{potassium}$. As hypokalemia and hyperkalemia both induce arrhythmia, in a scenario of high arrhythmia risk, the fusion of their value and feature identity embeddings should ideally be the same to represent the common phenotype. For MuFuse, this can be easily done by masking inconsistent entries of $\mathbf{e}_{v(low)}$ and $\mathbf{e}_{v(high)}$ by the learned $\mathbf{e}_{potassium}$ during the element-wise multiplication while keeping $\mathbf{e}_{v(low)}$ and $\mathbf{e}_{v(high)}$ different to represent other phenomena in the scenario. On the contrary, if we fuse them through addition, $\mathbf{e}_{v(low)}$ and $\mathbf{e}_{v(high)}$ must be the same to produce an identical fused representation, losing the flexibility.

