# OpenReview forum: "MedFuse: Multiplicative Embedding Fusion for Irregular Clinical Time Series"
_ICLR.cc/2026/Conference — ICLR 2026 Conference Withdrawn Submission_

### Official Review · Reviewer_8iZu · 2025-11-01

**Soundness:** 2
**Presentation:** 1
**Contribution:** 1
**Rating:** 2
**Confidence:** 4

**Summary:**

The paper proposes MedFuse, a framework for irregular clinical time series centered on the MuFuse (Multiplicative Embedding Fusion) module. MuFuse fuses value and feature embeddings through multiplicative modulation, preserving feature-specific information while modeling higher-order dependencies across features. Experiments on two datasets demonstrate the superior performance of MedFuse.

**Strengths:**

1. The studie problem is interesting and important. Especially irregular time series analysis is a pretty challenging domain for clinical data analysis.

2. The proposed approach achieve good results on 3 different tasks.

**Weaknesses:**

1. The paper’s contribution appears incremental. Representing each feature at each timestamp as an embedding follows prior work in SUMMIT [1], as the authors acknowledge. The main novelty is the multiplicative fusion of the feature identifier and the value embedding. As presented, this reads more as a heuristic than a method: the paper does not explain why multiplication is preferable to alternatives such as addition, concatenation, gating, attention, or bilinear pooling.

2. Empirically, the gains are unclear. In Table 1, MedFuse and SUMMIT achieve overlapping performance once the reported standard deviations are considered, suggesting no statistically meaningful improvement.

3. In addition, several figures and tables (e.g., Figure 1, Figure 3, Table 2) closely mirror SUMMIT’s presentation, which further blurs what is substantively new beyond [1].


[1]. Huang, Chun-Kai, Yi-Hsien Hsieh, Ta-Jung Chien, Li-Cheng Chien, Shao-Hua Sun, Tung-Hung Su, Jia-Horng Kao, and Che Lin. "Scalable Numerical Embeddings for Multivariate Time Series: Enhancing Healthcare Data Representation Learning." arXiv preprint arXiv:2405.16557 (2024).

**Questions:**

1. What is the rationale for choosing a multiplicative operation to fuse feature-identity and value embeddings? Please provide theoretical motivation or comparative experiments against common alternatives (addition, concatenation, gating/MLP, attention, bilinear).

2. Table 2 reports ablations only on PhysioNet 2012 (P12). Could you extend these ablations to other datasets or tasks to demonstrate robustness and generality?

3. What is the intended takeaway of Table 3? As written, the transfer results do not show clear gains, and performance when transferring from P12 to MIMIC-III appears to worsen.

---

### Official Review · Reviewer_EooK · 2025-11-08

**Soundness:** 1
**Presentation:** 2
**Contribution:** 1
**Rating:** 0
**Confidence:** 3

**Summary:**

This paper proposes MedFuse, a model incorporating irregular clinical time series. The core of the paper is a novel 'MuFuse embedding module' that performs value-conditioned multiplicative fusion. The multiplicative factors themselves are obtained via sigmoid function to supress outliers. The authors claim this approach better captures nonlinear feature-value interactions compared to existing fusion methods that are generally additive. They evaluate MedFuse on three clinical datasets (two public and one private) showing performance improvements over several baselines. The paper also includes a cross-dataset transfer experiments to support the idea of learning cohort-agnostic features.

**Strengths:**

The main contribution of fusion methods for irregular EHR data is an interesting and underexplored area. The paper generalizes the SUMMIT model, and the derivation is sound.
The formulation and mathematical notation are generally clear presented. The overall paper is easy to read. The experiments are not just on ICU mortality prediction datasets, but also carcinoma, suggesting some generalizability.
If the claims hold, the approach proposed in the paper can influence how numerical values are embedded in clinical models.

**Weaknesses:**

The paper heavily relies on the claim that multiplicative fusion enables "richer feature-value interactions" and "nonlinear modulation," but provides limited evidence about its benefit for clinical data.

The justification for multiplicative gating is also hand-wavy, and relevant only at the last layer. Since we have a transformer architecture with multiple layers, there is no reason to see why the relevant interaction cannot be learned under additive terms. Furthermore, there are multiple papers which have porposed incorporating multiplicative interactions. How is this work different?

The paper talks about irregularly sampled time series, but I do not see anything specific to that use case here. Any argument in the paper can be applied even to regularly sampled no-missing data series as well.

The improvements are minor, and are often not significant. This might not be as important, if the paper established that these differences are clinically meaningful. But I see no such evidence. Additionally all the results are based on only 2 datasets PhysioNet and MIMIC.  The third HCC data is entirely private with no reference.

Furthermore on Physionet, accuracy increases only marginally but the model deviation increased 4 times. This undermines the entire argument about the model capturing specific multiplicative interactions which others do not.

**Questions:**

What specific types of clinical patterns or relationships does your method capture that is not captured by other methods?

What is the value of Sec 5.3. The alternate parameterization was not experimented on nor theoretically analyzed, so this seems like unnecessary addendum to the paper.

As the paper also says, the transfer results are influenced more by dataset size rather than embedding quality. Moreover the descriptions says that after initial freezing, the embeddings is fine-tuned as well. Doesn't this undermine the claim of learning "reusable, cohort-agnostic embedding"?

Additionally multiplicative interactions baseline models can be added. See [1,2,3] and references there in

 Have you conducted any error analysis to identify specific patient subgroups or clinical scenarios where the improvement is most pronounced?

IMPORTANT: I was a last minute/emergency reviewer, so I have not had the chance to read this in detail. If I have misunderstood or missed anything, please bring it to my notice, and I will correct myself and revise my ratings.

Minor:
There are many duplicated references ( Song et al, Shukla and Marlin, Tipirneni and Reddy, etc.)

1 Multiplicative interactions and where to find them, Jayakumar et al
2 AdaDHP: Fine-Grained Fine-Tuning via Dual Hadamard Product and Adaptive Parameter Selection, Liu et al
3 SCINet: Time Series Modeling and Forecasting with Sample Convolution and Interaction

---

### Official Review · Reviewer_MULN · 2025-11-08

**Soundness:** 2
**Presentation:** 3
**Contribution:** 2
**Rating:** 2
**Confidence:** 5

**Summary:**

To effectively model numerical laboratory measurements, this paper proposes MedFuse to disentangle irregular clinical time series numerical values into value and feature embeddings.

**Strengths:**

- The motivation is clear.
- The problem is interesting.
- The paper is easy to read and understand.

**Weaknesses:**

- Insufficient experiments.
- Novelty is limited. It is a technique that disentangles numerical values into two parts. But this has been done by others like FT-Transformer and TabTransformer-like series models.

**Questions:**

- Novelty clarification. Please reclarify your novelty after surveying the Transformers for tabular data.
- More details about the standard lookup table are needed.
- The method is claimed to model the feature identity embeddings and value embeddings. Validation experiments or detailed interpretation are needed to support your claim.
- Why repeat each entry of $e_v$ for k times? There are other choices, like using an MLP (or other layers) to map the two embeddings into one shared embedding. Repeating k times does not make sense.
- The TabTransformer-like series models should also be included as baselines.
- The step in "Comparison of different partitioning factors k on P12" is too large. A fine-grained setting is needed.

---

### Note · Authors · 2025-11-13

**Comment:**

We sincerely thank all the reviewers for their detailed and constructive reviews. We have decided to withdraw this submission and thoroughly revise it by incorporating all the suggestions and addressing all the concerns and questions. We appreciate the reviewers' tremendous effort in helping us improve our submission.

**Withdrawal Confirmation:**

I have read and agree with the venue's withdrawal policy on behalf of myself and my co-authors.